# Effect of Curcumin Consumption on Inflammation and Oxidative Stress in Patients on Hemodialysis: A Literature Review

**DOI:** 10.3390/nu15102239

**Published:** 2023-05-09

**Authors:** Javiera D’andurain, Vanessa López, Migdalia Arazo-Rusindo, Caterina Tiscornia, Valeria Aicardi, Layla Simón, María Salomé Mariotti-Celis

**Affiliations:** 1Nutrition and Dietetic School, Facultad de Medicina, Universidad Finis Terrae, Pedro de Valdivia 1509, Santiago 7501015, Chilectiscornia@uft.cl (C.T.);; 2Institute of Nutrition and Food Technology, Escuela de Post Grado, Universidad de Chile, El Líbano 5524, Santiago 8331051, Chile

**Keywords:** chronic kidney disease, renal replacement therapies, inflammation, oxidative stress, curcumin, bioaccessibility

## Abstract

Advanced chronic kidney disease (CKD) stages lead to exacerbated inflammation and oxidative stress. Patients with CKD in stage 5 need renal hemodialysis (HD) to remove toxins and waste products. However, this renal replacement therapy is inefficient in controlling inflammation. Regular curcumin consumption has been shown to reduce inflammation and oxidative stress in subjects with chronic pathologies, suggesting that the daily intake of curcumin may alleviate these conditions in HD patients. This review analyzes the available scientific evidence regarding the effect of curcumin intake on oxidative stress and inflammation in HD patients, focusing on the mechanisms and consequences of HD and curcumin consumption. The inclusion of curcumin as a dietary therapeutic supplement in HD patients has shown to control the inflammation status. However, the optimal dose and oral vehicle for curcumin administration are yet to be determined. It is important to consider studies on curcumin bioaccessibility to design effective oral administration vehicles. This information will contribute to the achievement of future nutritional interventions that validate the efficacy of curcumin supplementation as part of diet therapy in HD.

## 1. Introduction

Chronic kidney disease (CKD), also called chronic kidney failure, is a general term for heterogeneous disorders affecting gradually the structure and function of the kidney [1]. CKD is a global health problem that affects more than 10% of the world population (843.6 million) [2]. The World Health Organization has described CKD as the fourteenth leading cause of death in the world.

CKD has five stages, defined according to the glomerular filtration rate (GFR). In stages 1 and 2, dietary therapeutic management focuses on reducing CKD progression, cardiovascular risk (CVR) and high blood pressure. In stage 3 onwards, control of the comorbidities inherent to CKD (anemia, bone alteration, protein-energy wasting (PEW), water and electrolyte imbalance) is added. In stage 5, patients who cannot manage to halt the progression of the pathology need conservative non-dialytic management or renal replacement therapies (RRT: HD, peritoneal dialysis (PD) or renal transplantation) [3].

CKD is associated with inflammation and oxidative stress. Many factors promote the inflammatory state in CKD such as increased production of proinflammatory cytokines, acidosis, chronic and recurrent infections, intestinal dysbiosis and altered adipose tissue metabolism. Oxidative stress in CKD is a consequence of the increment in reactive oxygen species (ROS) production and the decrease in antioxidant capacity. Moreover, oxidative stress is promoted by inflammation. In addition, oxidative stress promotes the inflammatory response by increasing pro-inflammatory mediators (e.g., NF-κB) [4]. Furthermore, inflammation and oxidative stress are increased in patients on HD. HD is the process by which toxins and waste products diffuse from the blood into a countercurrent dialysate through a semipermeable membrane [5]. CKD patients on HD have an increment in NF-κB and a decrease in antioxidant proteins such as nuclear factor E2-related factor 2 (Nrf2) and NADPH: quinoneoxidorectase 1 (NQO1) [6].

Evidence indicates that more patients with CKD will require frequent RRT, e.g., maintenance HD is recommended at least three times per week with a minimum efficiency of 1.4 Kt/V per treatment (the Kt/V parameter is the ratio of urea clearance (K) multiplied by dialysis time (t) to the volume of water in the body (V) [7,8,9,10].

As a consequence of population aging and an increasing prevalence of diabetes and hypertension in both developed and developing countries, the worldwide prevalence of HD patients is expected to rise by over 60% [10,11,12]. The health burden associated with patients on HD represents a critical problem for the global healthcare system, considering the increasing rate of patients requiring HD, the onset of the pathology and the life expectancy of HD patients [13]. The magnitude and seriousness of the situation described reflect the need to develop complementary therapies to the pharmacological and instrumental treatments currently applied in patients on HD in order to improve their life quality. Within the framework of traditional diet therapy, several supplements based on bioactive compounds improve the status of HD patients, controlling oxidative stress and inflammation [14,15,16,17,18,19]. Among the bioactive compounds with antioxidant and anti-inflammatory activity, curcumin stands out [20,21,22].

Curcumin is the main active constituent of turmeric, and it is characterized by containing phenolic groups which confer antioxidant and anti-inflammatory effects (Figure 1) [23]. For that reason, the use of curcumin has emerged as a potential therapy to reduce the inflammation and oxidative stress in CKD patients [24]. In fact, curcumin supplementation in CKD patients on HD reduces the expression of inflammatory markers such as NF-κB, C-reactive protein (CRP) [23] and tumor necrosis factor-alpha (TNF-α) [25,26].

Despite the potential benefits of curcumin, its high lipophilicity can result in low absorption and rapid metabolism and systemic elimination when consumed in an inadequate matrix [27,28]. The bioavailability of a bioactive compound, defined as its intestinal absorption and biological effect at the site of action, is largely determined by its bioaccessibility from the food matrix in which it is consumed. This association has led to the development of controlled delivery oral systems that modulate the release of bioactive compounds during gastrointestinal digestion to increase their bioavailability [29]. This literature review evaluates the intake of curcumin as a potential complement to traditional diet therapy in patients on HD. It analyzes (i) the mechanisms and consequences of HD in the organism, (ii) the mechanisms of action and effectiveness of curcumin consumption in controlling the inflammation and oxidative stress in HD patients and (iii) the effect of curcumin bioaccessibility on its effectiveness as an anti-inflammatory and antioxidant agent.

## 2. Methodology

A search for information was performed in databases such as PubMed, Scielo, Google Scholar and Scopus of the Finis Terrae University Library. A total of 89 articles were considered, of which 19 (~23%) were published in the last 5 years (from 2018 to date), and 62 (~69%) were published in the last 10 years (from 2013 to date). As a search strategy, the following terms were used in different combinations (one, various or all of these terms) in the title: “chronic kidney disease”, “hemodialysis”, “inflammation”, “oxidative stress”, “curcumin” and “bioavailability”.

### Selection Procedure

The titles of the publications were analyzed to assess whether the papers were related to the objectives of this review. Articles published in English and Spanish related to the association between curcumin consumption and its effect on inflammation and oxidative stress in HD patients were considered as inclusion criteria. Both observational (case-control and prospective longitudinal studies) and experimental (clinical trial studies) studies were included, as well as systematic reviews and meta-analyses. On the other hand, exclusion criteria were considered: (1) articles with subjects suffering from another inflammatory or oxidative disease or condition, (2) studies with persons with a habitual intake of curcumin as dietary therapeutic management of a previous clinical condition, (3) articles with patients with renal transplantation scheduled within the study period, (4) studies in which curcumin was not the main exposure and (5) patients with use of antioxidant or anti-inflammatory supplements in the last 3 months. Finally, duplicate articles were eliminated.

## 3. Results

### 3.1. Mechanism and Consequences of HD in the Organism

CKD is primarily caused by the growing elderly population and the increasing occurrence of diabetes and hypertension worldwide. Research predicts that by 2030, the number of individuals globally requiring RRT will rise to 5.4 million, with the highest growth rates anticipated in Asia and Latin America [11]. In these regions, rapid urbanization and globalization have led to overlapping disease burdens, resulting in high and growing rates of infectious diseases, diabetes and hypertension. Furthermore, in developing countries, CKD is exacerbated by environmental pollution, pesticide exposure, analgesic abuse, the use of herbal medications and the ingestion of unregulated food additives [11,12]. Dialysis remains the most prevalent RRT in end-stage CKD (stage 5), and HD is the most commonly applied modality globally [10].

Dialysis is a purification technique that filters toxic substances accumulated in the serum and water through a countercurrent extracorporeal circuit, consisting of a dialyzing filter and machine for the transport of blood and dialysate [30]. This technique promotes inflammation and oxidative stress in patients (Figure 2). This situation is exacerbated (hyperinflammation) using non-biocompatible membranes and non-ultrapure water. Under these conditions, the capacity to remove high molecular weight molecules is diminished, and possible contamination of the dialysis fluid may occur. As a consequence, the transfer of bacteria into the blood could trigger the activation of mononuclear cells and the production of proinflammatory cytokines, such as interleukin 1 (IL-1) or TNF-α, which could promote inflammation, oxidative stress and deterioration of the immune system [31].

The inflammatory response generates endothelial damage, overproduction of reactive oxygen species (ROS), lipid peroxidation and decrease in the production of nitric oxide (NO). As a consequence, there is an increased oxidation in HD patients, resulting in a loss of balance between antioxidant compounds and free radical levels that contribute to cellular and tissue damage. Together with this, HD increases the concentration of hypercatabolic hormones, meanwhile decreasing the level of anabolic hormones [32].

On the other hand, the induced microinflammation contributes to the onset of PEW, a pathological condition characterized by a reduction in protein stores and energy reserves, leading to a decline in muscle mass and adipose tissue, respectively [33,34,35]. This syndrome commonly manifests itself in 30–60% of dialysis patients, due to the basal proinflammatory state of this therapy [34]. In addition, low intake and/or insufficient absorption of nutrients has been observed in HD patients, which is associated with age, uremia, concomitant pathologies and the alteration of hormones such as leptin and ghrelin [36]. Likewise, there is a loss of nutrients resulting in a notable negative energy balance due to increased protein catabolism that leads to muscle atrophy mediated by the activation of the ubiquitin–proteasome proteolytic system and caspases. Moreover, the dialytic technique causes a constant loss of amino acids, reducing their bioavailability for muscle protein synthesis [34,36,37,38]. For that reason, the nutritional approach through diet therapy is crucial in the evolution of this pathology and in the management of the nutritional status of HD patients.

Regarding nutritional requirements, energy requirements are of great relevance in HD patients and differ from those of healthy persons. In healthy patients, 25 to 30 kcal/kg/d is indicated, while in patients with CKD at any stage and metabolically stable, an intake of 25 to 35 kcal/kg/d is recommended to maintain a normal nutritional status. This recommendation may vary depending on age, gender, level of physical activity, body composition, secondary diseases or presence of inflammation [39,40].

Optimal daily protein requirements in healthy patients are prescribed from 0.8 to 1.0 g/kg/day, while in metabolically stable CKD patients (stages 1 to 4), the level of protein intake can be safely reduced to 0.55–0.60 g/kg/day. In addition, a further reduction in protein intake to 0.3–0.4 g protein/kg/day can be achieved through the consumption of keto and aminoacidic tablets, which ensure a sufficient balance of essential amino acids provided mainly by proteins of animal origin. In the case of patients with CKD in HD who are metabolically stable, a higher protein intake (1.0 to 1.2 g/kg/day) is recommended because their clinical situation (PEW) entails hyperprotein and hypercaloric requirements, which may be even higher if the patient is also diabetic [39,41]. Lipid and carbohydrate requirements remain the same in both groups [33,39]. Micronutrient requirements should be individualized and restricted (phosphorus, potassium, sodium) depending on the clinical situation and metabolic conditions. Multivitamins should be indicated considering water-soluble, fat-soluble vitamins and trace elements for possible deficiencies [39].

Although the nutritional requirements are known, around 29% of women and 50% of men on HD have malnutrition or excessive weight. Moreover, the diets of HD patients are deficient in protein, energy, dietary fiber, vitamins B_1_, D and C, folates and Ca and Mg [42] which aggravates the general status of patients. Recommendations of suitable diets and supplementations with vitamins, minerals and alternative therapeutic compounds are mandatory to alleviate patient suffering.

### 3.2. Mechanism of Action and Effect of Curcumin Consumption on the Control of Oxidative and Inflammatory Parameters in HD Patients

CKD in HD is a highly complex pathology, for that reason, it is necessary to supplement diets with several micronutrients (vitamins and minerals) and bioactive compounds (polyunsaturated fatty acids and polyphenols) [14,15,17,18,19,33,34,43]. In this sense, the intake of curcumin, a polyphenol of natural origin, stands out as an alternative for the control of various oxidative and inflammatory parameters associated with renal damage. Structurally, curcumin has a diketone hydrocarbon skeleton that unites two phenolic rings (Figure 1), which provide antioxidant and anti-inflammatory properties [29,33].

Studies indicate that the hydroxyl groups existing in the phenolic rings of curcumin regulate the expression of antioxidant enzymes such as superoxide dismutase (SOD), catalase (CAT), glutathione-S-transferase (GST), glutathione peroxidase (GPX) and heme oxygenase 1 (HO-1) (Figure 3). The phenolic rings are suggested to scavenge and reduce ROS such as superoxide, hydrogen peroxide, hydroxyl, peroxyl and NO radicals [28,44,45,46,47,48] (Figure 3). In addition, through ketone groups, curcumin exerts an antioxidant role in protecting cellular DNA from free radicals. As a consequence of this antioxidant potential, curcumin decreases inflammatory markers such as TNF-α, CRP and IL-1, serum creatinine and urea nitrogen (BUN) levels. Therefore, curcumin improves GFR and attenuates renal dysfunction [21,28,49,50] (Figure 3).

Along with previous evidence, curcumin has an antioxidant role by reducing lipid peroxidation and anti-inflammatory activity by binding to inflammatory molecules such as TNF-α, α1-human acidic α-glycoprotein (α1-GA), myeloid differentiation protein 2 (MD-2), cyclooxygenase-1 (COX-1) and 2 (COX-2) and the lipoxygenase (LOX) pathway [29,48] (Figure 2). Curcumin inhibits the COX and LOX pathways and, consequently, the metabolism of arachidonic acid, thereby affecting the synthesis of eicosanoids such as prostaglandins (PG), thromboxanes (TXA) and leukotrienes (LT), preventing the development of inflammatory processes and platelet aggregation [50,51]. Additionally, curcumin suppresses TNF-α and interleukin expression by inhibiting NF-κB. Moreover, curcumin suppresses STAT3, Wnt, PI3K and mitogen-activated protein kinase (MAPK)-dependent signaling pathways and modulates TGF-β expression and activity. Downstream of these events, curcumin exerts an anti-inflammatory role by decreasing proinflammatory cytokines [3,26,50,52].

At the preclinical level, curcumin administration exerts a protective effect on animal models for streptozotocin (STZ)-induced diabetic nephropathy [53,54,55,56], cisplatin-induced nephrotoxicity [57,58,59], 5/6 nephrectomy [60] and renal ischemia-reperfusion (I/R) injury [61]. In all of these studies, curcumin reduces oxidative and inflammatory markers regardless of the dose, the type of vehicle (pure curcumin, Arabic gum suspensions, carboxymethylcellulose (CMC), PBS, olive oil, nanoparticles) and the route of administration (feeding, oral gavage, intravenous, intraperitoneal) (Table 1). For example, in rats with STZ-induced diabetic nephropathy, chronic consumption of curcumin (orally and by oral gavage) attenuated renal dysfunction [54], macrophage infiltration [55] and fibrosis [56], probably due to its antioxidant properties. Likewise, in rats with cisplatin-induced nephrotoxicity, oral [57,58] and intraperitoneal [59] consumption of curcumin presented a protective effect, reduced the levels of oxidative and inflammatory markers and could be used as a potential coadjutant treatment. In 5/6 nephrectomized rats, curcumin attenuated oxidative stress, inflammation and renal fibrosis by modulating the Nrf2-Keap1 pathway [60], suggesting that it is a potential treatment for CKD. Similarly, rats with I/R injury subjected to daily oral administration of curcumin showed a reduction in urea and creatinine levels, an increase in glutathione (GSH), a decrease in malondialdehyde (MDA) and, consequently, reduced oxidative stress [46]. In addition, curcumin modulated the inflammatory response induced by I/R in the rat kidney through the activation of the JAK2/STAT3 signaling pathway [61].

Regarding clinical studies (Table 2), HD patients who received 2.5 g of turmeric powder (95% curcumin) diluted in 100 mL of orange and carrot juice after each dialysis session/week for 3 months experienced a decrease in NF-kB and CRP [23]. Another intervention considered the effect of curcumin on plasma levels of uremic toxins, such as indoxyl sulfate, p cresyl sulfate (pCS) and 3-indoleacetic acid, in HD patients. The results showed a significant decrease in pCS [63]. Similarly, the daily intake of 120 mg of nano-curcumin supplementation, delivered as three soft gelatin capsules with breakfast, lunch and dinner for 12 weeks, resulted in a significant decrease in inflammation markers (CRP, ICAM-1 and vascular cell adhesion protein 1 (VCAM-1)) in HD patients [64].

Likewise, the intake of turmeric capsules containing 22.1 mg of curcumin (three times a day for eight weeks) resulted in a significant decrease in CRP, IL-6 and TNF-α levels [49], as well as increased levels of antioxidants such as GPX, glutathione reductase (GR) and CAT [22,65] in HD patients, without adverse effects (Table 2). This suggests that curcumin administration may have a coadjutant role in reducing inflammatory indicators and pCS at the intestinal level and increasing the levels of certain endogenous antioxidants in HD patients. In this regard, it would be inaccurate to assert that the doses used in these studies are adequate since they did not consider the body weight of HD patients. Moreover, these patients experience weight fluctuations before and after HD, making it challenging to determine the appropriate dosage.

### 3.3. Effect of Curcumin Bioaccessibility on Its Antioxidant and Anti-Inflammatory Effectiveness

Previously discussed clinical trials with curcumin have shown effectiveness in enhancing different inflammatory and antioxidant biomarkers in CKD patients. However, some of the evaluated doses could be higher than the acceptable daily intake established for curcumin (3 mg/day kg bw) by the European Food Safety Authority [66].

The hydrophobic nature of curcumin can limit its antioxidant and anti-inflammatory activity when administered orally. The insufficient solubility of orally administered curcumin results in low bioaccessibility during digestion, and as a consequence, there is a decrease in its availability [67]. Furthermore, curcumin undergoes accelerated metabolism and is rapidly eliminated systemically, primarily through the feces [68,69,70,71].

To overcome these limitations, several technologies have been developed to enhance the bioaccessibility and bioavailability of curcumin at the intestinal level, thereby increasing its therapeutic potential. These techniques are based on triglycerides and tensoactive mixtures that physically isolate curcumin from gastrointestinal fluids and protect it from chemical degradation. By utilizing a digestible lipid phase, such as a triglyceride, fatty acids and monoglycerides are produced and incorporated into mixed micelles, which solubilize curcumin within their hydrophobic core, thereby improving its bioaccessibility [72]. Among these formulations are oil-in-water emulsions [73,74], microemulsions [75], solid dispersions [76], solid lipid nanoparticles [77], liposomes [78], lipid solutions and self-emulsifying drug and bioactive delivery systems (SEDDS) [79].

Emulsion-based delivery systems, particularly oil-in-water (O/W) emulsions, are very convenient for solubilizing lipophilic compounds such as curcumin. In these colloidal systems, curcumin is first solubilized within the oily phase, and then this phase is homogenized within the aqueous phase containing the emulsifier [74,77]. O/W emulsions consist of small lipid droplets coated with the emulsifier and dispersed into an aqueous phase. O/W emulsions are thermodynamically unstable systems that require high-energy emulsification methods (e.g., high-pressure homogenization, sonication, microfluidization, among others) for stabilization. A conventional O/W emulsion consists of lipid droplets with diameters greater than 200 nm. When the droplets have a diameter between 20 and 200 nm, they are called nanoemulsions. In contrast, microemulsions vary in their thermodynamic stability and can therefore be formed spontaneously by mixing their components [80].

Several studies have suggested that the use of emulsion-based delivery systems increases curcumin bioaccessibility, due to the formation of mixed micelles during lipid digestion, which in turn increases the solubility of curcumin in the intestinal fluid [32,81,82,83]. The composition and structure of these systems are specifically designed to improve curcumin bioavailability by increasing its bioaccessibility in the gastrointestinal tract [74]. For example, in emulsions and nanoemulsions, the type of lipid significantly affects the in vitro intestinal digestion of curcumin. The curcumin bioaccessibility increases in emulsions (58%) and nanoemulsions (59%), which contain medium-chain fatty acids possibly because of their higher capacity to form mixed micelles during digestion [49].

On the other hand, liposomes are vesicles that present one or more bilayers, generally formed by phospholipids, which allow them to enclose both lipid-soluble (e.g., curcumin) and water-soluble substances [78,84]. Liposomes with a variable diameter (20 to 400 nm) are obtained by forming a lipid film that is resuspended in a specific aqueous system and extruded or sonicated. Other in vitro digestion studies have concluded that curcumin bioaccessibility is increased (1.7-fold concerning the free compound) when it is contained in liposomes [76,85,86]. These results could explain why curcumin-loaded liposomes showed 4-fold higher absorption and 2.4-fold higher plasma antioxidant activity compared to free curcumin administered orally to rats (100 mg/day/kg by gavage) [84].

SEDDS are mixtures of lipids, surfactants and cosolvents that form fine O/W emulsions once dispersed into an aqueous medium under gentle agitation. In this sense, gastrointestinal tract fluids and motility provide the necessary medium and the agitation for the self-emulsification of these formulations [27]. In this way, SEDDS form loaded emulsion droplets that increase the solubility and absorption of orally consumed hydrophobic compounds, such as curcumin [79]. SEDDS significantly increase the curcumin solubility (70–100%) compared to the free and pure compound at both neutral pH (6.8) and acidic conditions (0.1 N HCl) [79,87]. Results of acute preclinical studies (24 h, 25–100 mg/kg day) have shown that oral administration of curcumin-loaded SEDDS significantly increases its plasma absorption (~50–94%) by binding curcumin with the lipid fraction of the emulsion and thereby increasing its bioaccessibility [79,87,88].

Another suggested approach to improve the bioavailability of curcumin involves the use of bioenhancers, such as piperine. Piperine is known to enhance curcumin absorption and tissue uptake, while also reducing curcumin hepatic metabolism. Once absorbed, curcumin is metabolized within the epithelial cells and/or effluxed back into the intestinal lumen. This efflux is thought to occur due to the presence of efflux transporters within the intestinal cell membranes. Piperine from black pepper, as well as certain catechins from green tea, are able to inhibit these transporters, thus increasing the amount of curcumin that remains in the body [89]. A preclinical study administrated an orally delivered powder of curcumin (2 g/kg/day) and piperine (20 mg/kg/day) and concluded that piperine significantly increases the serum concentration of curcumin, resulting in a 154% improvement in its relative bioavailability. Similarly, a clinical study conducted with the combined oral administration of curcumin and piperine dissolved in water (2 g of curcumin powder, 20 mg of piperine and 150 mL of water). The study found that piperine significantly increased the serum concentration of curcumin, resulting in a relative bioavailability increase of 2000% [47].

## 4. Conclusions

HD significantly increases the oxidative and inflammatory parameters in patients undergoing RRT. The regular intake of natural antioxidants, as a complement to dietary therapy for CKD patients, has been shown to effectively control these biomarkers, thereby contributing to reducing the adverse effects of HD. One of the most promising bioactive compounds is curcumin, a major polyphenol in turmeric that has been successfully evaluated in the treatment of several chronic pathologies, including CKD.

Several mechanisms of action associated with the antioxidant role of curcumin have been proposed as responsible for its anti-inflammatory activity. This has been evidenced in preclinical and clinical studies where regular intake of variable doses of curcumin has been effective in controlling oxidative and inflammatory parameters. However, the most effective oral administration vehicle for curcumin has not yet been defined. Due to its physicochemical properties, the bioaccessibility and bioavailability of this compound are low. In this regard, several oral systems of controlled delivery of curcumin have been evaluated, among which liposomes and SEDDS stand out as the best at the preclinical level in terms of bioaccessibility and absorption of the bioactive compound. Likewise, oral administration of free curcumin together with adjuvants that decrease its hepatic metabolism, such as piperine, has increased its bioavailability in both rats and humans.

This literature review analyzed the available preclinical and clinical studies evaluating the antioxidant and anti-inflammatory effects of curcumin and its association with CKD. The background information gathered leads to the conclusion that curcumin consumption can be successfully integrated as a dietary therapeutic adjuvant in HD patients.

Future research should aim to increase scientific evidence by conducting nutritional interventions that incorporate curcumin into the basal diet therapy for HD patients. In this regard, it is critical that these studies include anthropometric information about patients to standardize the curcumin dosage according to its acceptable daily intake. Moreover, it is essential to consider the bioaccessibility of curcumin in the rational design of the administration vehicle, particularly for oral consumption, to enhance its bioavailability and ensure its effectiveness.

## Figures and Tables

**Figure 1 nutrients-15-02239-f001:**
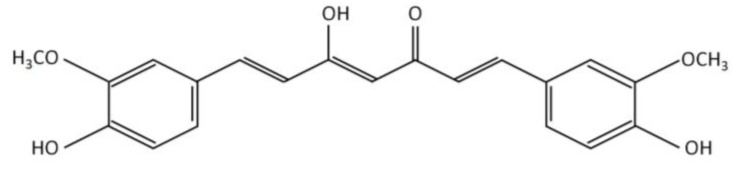
Chemical structure of curcumin.

**Figure 2 nutrients-15-02239-f002:**
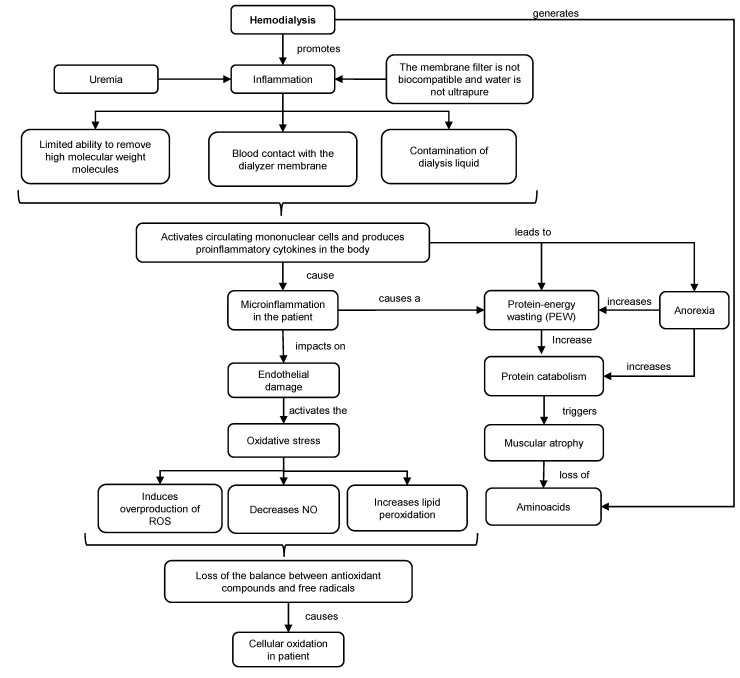
Mechanism of inflammation and oxidative stress in HD. ROS: reactive oxygen species. NO: nitric oxide.

**Figure 3 nutrients-15-02239-f003:**
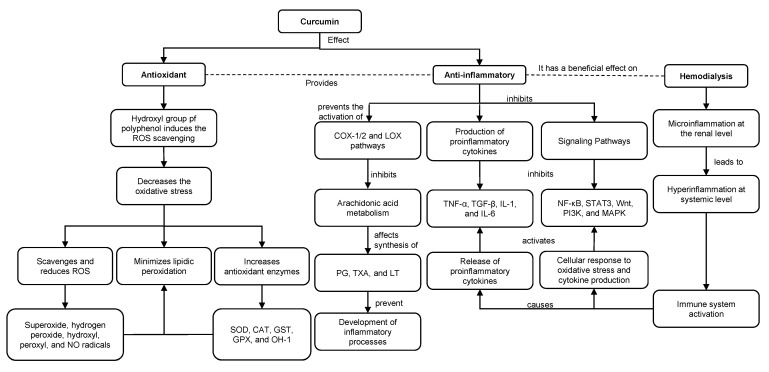
Antioxidant and anti-inflammatory roles of curcumin in HD. ROS: reactive oxygen species. NO: nitric oxide. SOD: superoxide dismutase. CAT: catalase. GST: glutathione-S-transferase. GPX: glutathione peroxidase. HO-1: heme oxygenase 1. COX-1: cyclooxygenase-1. COX-2: cyclooxygenase-2. LOX: lipoxygenase. PG: prostaglandins. TXA: thromboxanes. LT: leukotrienes.

**Table 1 nutrients-15-02239-t001:** Preclinical studies on curcumin ingestion in CKD.

Objective	Vehicle	Route of Administration	Dosage	Variable Response	Results	Conclusions	Reference
To explore the effect of curcumin consumption on experimental nephrotoxicity and oxidative stress induced by cisplatin	Emulsion with CMC	Oral	15, 30, 60 mg/kg/d, 1 dose 2 days before and 3 days after cisplatin injection (CPT)	BUN, creatinine, MDA, GSD, SOD, CAT and TNF-α	(↓) the increase in BUN, creatinine, MDA, GSD, SOD, CAT, and TNF-α caused by CPT Improvement in morphological damage to the kidney	Curcumin has a protective effect on cisplatin-induced experimental nephrotoxicity due to its antioxidant and anti-inflammatory activity	[58]
To determine the protective effect of curcumin on CPT-induced nephrotoxicity	Olive oil	Intraperitoneal tube	100 mg/kg/d, 1 dose	BUN, creatinine, TNF-α, MCP-1, ICAM-1 mRNA	(↓) the increase in BUN, creatinine, serum TNF-α, renal TNF-α, renal MCP-1 and ICAM-1 mRNA caused by CPT injection	Curcumin has a protective effect on cisplatin-induced experimental nephrotoxicity due to its antioxidant and anti-inflammatory activity	[59]
To explore the effect of curcumin consumption on experimental nephrotoxicity and oxidative stress induced by CPT	Nanoparticles	Oral	50 mg/kg/d; 14 days	BUN, creatinine, MDA, NO, GSH and TNF-α	(↓) the increase in BUN, creatinine, MDA, NO and TNF-α caused by CPT Prevents the (↓) of GSH caused by CPT Improvement in morphological damage to the kidney	Curcumin has a protective effect on cisplatin-induced experimental nephrotoxicity due to its antioxidant and anti-inflammatory activity	[57]
To evaluate the effect of curcumin on streptozotocin (STZ)-induced diabetic nephropathy (DN) in rats	Emulsion with CMC	Oral	15–30 mg/kg/d; 2 weeks	Creatinine, BUN, albumin, MDA, GSH, CAT, SOD	(↓) creatinine and BUN and MDA (↑) GSH, CAT, SOD in rats DN	Chronic treatment with curcumin attenuates renal dysfunction and oxidative stress in rats (DN)	[54]
To investigate whether curcumin reduces macrophage infiltration in STZ-induced DN rats	Emulsion with gum Arabic	Oral	100 mg/kg/d; 2 weeks	Creatinine, BUN, NF-KB, TNF-α, IL-1, ICAM-1, TGF-β.	(↓) creatinine, BUN (↓) of NF-KB, TNF-α, IL-1, ICAM-1, TGF-β, 1 expression in DN rats (↓) of macrophage infiltration in the kidneys of DN rats	Curcumin reduces macrophage infiltration in kidneys of STZ-induced DN rats by inhibiting NF-KB activation and proinflammatory cytokines	[55]
To evaluate whether curcumin arrests the development of STZ-induced DN in rats, by inhibiting PKC-α and PKC-β1 activity and ERK1/2 pathway	Emulsion with gum Arabic	Oral probe	100 mg/kg/d; 8 weeks	Creatinine, BUN, MDA, GSH, PKC-α and PKC-β1	(↓) creatinine, BUN, and MDA β1 in DN rats (↑) GSH. β1 in DN rats (↓) of PKC-α and PKC-β1 in DN rats	Curcumin protects against the development of DN, which involves a dual blockade of PKC-α and PKC-β1 as well as the downstream pathway, ERK1/2. Probably due to its antioxidant properties, curcumin has an antifibrotic effect	[56]
To evaluate the effectiveness of curcumin relative to enalapril on chronic kidney disease (CKD) in 5⁄6 nephrectomized rats	Emulsion with gum Arabic	Oral probe	75 mg/kg/d; 8 weeks	Creatinine, BUN, TNF-α, IL-1	Curcumin was as effective as enalapril for (↓) increased BUN and creatinine (↓) levels of TNF-α and IL-1	Curcumin was as effective as enalapril for (↓) increased BUN and creatinine (↓) levels of TNF-α and IL-1	[62]
To evaluate whether curcumin, by increasing Nrf2 expression, could reduce oxidative stress, inflammation and renal fibrosis in 5⁄6 nephrectomized rats	Emulsion with gum Arabic	Oral probe	75 mg/kg/d; 8 weeks	Creatinine, BUN, MDA, GSH, COX-2, TNF-α, TGF-β, HO-1, Keap1, Nrf2, NF-KB	(↓) creatinine and BUN (↓) MDA (↑) GSH (↓) COX-2 (↓) TNF-α and TGF-β (↑) HO-1 * (↓) Keap1 * (↑) Nrf2 * (↓) NF-KB	Curcumin effectively attenuates oxidative stress, inflammation and renal fibrosis by modulating the Nrf2-Keap1 pathway, suggesting it has promising potential for the treatment of CKD	[60]
To evaluate the association between the antioxidant and anti-inflammatory effect of curcumin consumption and its action on ischemia-reperfusion injury (I/R)	Emulsion with CMC	Oral	200 mg/kg/d; 7 days	SOD, GSH, MDA, protein carbonyl (PC) and NO	(↑) GSH in serum (↓) MDA, NO and PC in serum and tissue Near normal kidney morphology	Consumption of curcumin protects the kidneys against I/R due to its antioxidant effect	[46]
To explore the effects and potential mechanisms of curcumin in modulating I/R-induced inflammatory response in rat kidney.	Emulsion with CMC	Intraperitoneal injection	60 mg/kg, 1 dose	Creatinine, BUN, IL-8, TNF-α and IL-6 in serum; mRNA level of IL-8, TNF-α and IL-6 in kidney, expression of JAK2, p-JAK2, STAT3, p-STAT3, p65 and p-p65 in the kidney	(↓) Creatinine, BUN, IL-8, TNF-α, IL-6 and p-p65 expression. (↑) the expression of p-JAK2 and p-STAT3. Attenuate pathological kidney injury. No effect on the expression of JAK2, STAT3 and p65	The protective effect of curcumin on I/R injury is associated with NF-KB-mediated suppression of inflammation through activation of the JAK2/STAT3 signaling pathway	[61]

↑ (increase); ↓ (decrease); * (*p* > 0.05).

**Table 2 nutrients-15-02239-t002:** Effect of curcumin consumption in CKD patients.

Objective	Vehicle	Route of Administration	Dosage	Variable Response	Results	Conclusions	Reference
To evaluate the effect of curcumin juice on the expression of inflammatory markers in HD patients	Oral juice	2.5 g of turmeric + 12 g of carrot + 100 mL orange juice	3 times per week after HD	NF-kB and PCR	(↓) NF-kB and PCR	Oral curcumin supplementation has an anti-inflammatory effect in HD	[23]
To evaluate the effect of curcumin supplementation on plasma levels of uremic toxins in HD patients	Oral juice	2.5 g of turmeric + 12 g of carrot + 100 mL orange juice	3 times per week after HD	IS, pCS, IAA	(↓) pCS but not in IS and IAA	Oral curcumin supplementation reduces plasma pCS levels in HD patients suggesting a modulation of the gut microbiota due to decreased production of uremic toxins	[63]
To evaluate the effect of nano curcumin supplementation on inflammation in HD patients	Nanocapsule	1 capsule = 40 mg curcumin	3 capsules daily	PCR, ICAM-1, VCAM-1	(↓) PCR and VCAM-1	Nanocurcumin shows beneficial effects in reducing inflammation in HD patients	[64]
To evaluate the effect of turmeric on markers of oxidative stress in HD patients	Capsule	1 capsule = 500 mg turmeric = 22.1 mg curcumin	3 capsules daily	MDA, GPX, GR, CAT	(↓) MDA. (↑) GPX, GR and CAT	Curcumin significantly attenuates oxidative species and increases antioxidant markers in CKD	[22]
To evaluate the effects of turmeric on the reduction of inflammatory markers in HD patients	Capsule	1 capsule = 500 mg turmeric = 21.1 mg curcumin	3 capsules daily	BUN, creatinine, IL-6, TNF-α, and CRP	(↓) CRP, IL-6 and TNF-α in HD patients	Curcumin capsule ingestion has an anti-inflammatory effect in HD patients	[49]
Investigating the effect of curcumin and ginger on oxidative modulation in CKD patients	Capsule	1 capsule = 500 mg	3 capsules daily	MDA, GPX, GR, CAT y SOD	(↓) MDA in patients with CKD(↑) GPX and CAT in CKD patients	Curcumin and ginger ingestion showed an increase in antioxidant markers in patients with chronic renal failure	[65]

↑ (increase); ↓ (decrease).

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
