# Peer review of "Effect of Curcumin Consumption on Inflammation and Oxidative Stress in Patients on Hemodialysis: A Literature Review"

_nutrients, 2023, doi:10.3390/nu15102239_

Round 1

Reviewer 1 Report

The authors has presented review titled "EFFECT OF THE CURCUMIN CONSUMPTION ON INFLAMMATION AND OXIDATIVE STRESS OF PATIENTS ON HEMODIALYSIS: A LITERATURE REVIEW". 

This review describe the mechanism of action and rolÄ™ of curcumin in aspect of patients with renal fakturÄ™, including HD patients. The subject of this review is extremely important since there are more and more patients who suffer from renal disfunction. Renal transplantation is currently the best treatment for this group of patient. HD are necessary for these patients, however, it is commonly known that HD leads to multiorgan disfunction,including heart.  Every section of the current article is ckearly described, with proper literature.

Conlusion is also provided and as far as I am concerned is understable.

Author Response

Reviewer 1 did not performed any comments or suggestions to include in the corrected version of the document. 

Please see the attachment, we decided to include the responses point by point to your comments (Editor comments)

Reviewer 2 Report

The manuscript by D’andurian et al. discusses in a good way an interesting topic. The review is clear and well-written. I suggest its publication in Nutrients after some minor revisions.

Introduction:

First line: abbreviations should be defined the first time they appear in the main text, even if present in the abstract. Please define CKD.

Same for HD, some sentences later.

Methodology:

We considered 76 articles published in the last 5 years (or from 2018 to date) and included in the title the terms: chronic kidney disease, hemodialysis, inflammation, oxidative stress, curcumin, and bioavailability.” Do they include just one/more than one/all these terms? Clarify.

Results:

“It is expected that by 2030, the use of RRT worldwide will reach 5.4 million people, with the highest growth projected in Asia.” Authors should discuss, here or in the introduction, which are the reasons behind this expected increase in patients requiring RRT. Also in the introduction this prediction is reported without an explanation.

Structurally, curcumin has a diketone hydrocarbon skeleton that unites two phenolic rings, which provide antioxidant and anti-inflammatory properties [25,38]”. Do the author think it might be useful to add curcumin chemical structure?

Please check the “Reference” columns in both tables. Reference numbers must be corrected.

“Likewise, the intake of turmeric capsules containing 22.1 mg of curcumin, three times a day for eight weeks), resulted in a significant decrease in CRP, IL-6, and TNF-α levels”. A round bracket is missing.

In Section (iii) the authors describe how the hydrophobic nature of curcumin limits its antioxidant and anti-inflammatory activity when administered orally, then describing different approaches to increase its bioavailability. However, in the previous section and in table 2 they report many examples of clinical studies where the oral administration of curcumin has shown anti-inflammatory and antioxidant positive effects. What is the reason behind the need of this optimized administration approaches? Are the required dosages (60-1500 mg/day) somewhat problematic? The authors should better clarify this point.
